# Every Mistake Counts: Spatial and Temporal Beliefs for Mistake Detection in Assembly Tasks

## Abstract

Assembly tasks, as an integral part of daily routines and activities, involve a series of sequential steps that are prone to error. This paper proposes a novel method for identifying ordering mistakes in assembly tasks based on knowledge-grounded beliefs. The beliefs comprise spatial and temporal aspects, each serving a unique role. Spatial beliefs capture the structural relationships among assembly components, indicating their topological feasibility. Temporal beliefs model the action preconditions and enforce sequencing constraints. Furthermore, we introduce a learning algorithm that dynamically updates and augments the belief sets in an online manner. To evaluate our approach, we first test its ability to deduce the predefined rules using synthetic data from industry assembly, and then apply it to a real-world dataset, enhanced with a new collection of annotations providing part information. We demonstrate our framework achieves superior performance in detecting ordering mistakes under both synthetic and real-world settings.

## 1 Introduction

In assembly procedures, components or parts are brought together in a precise and sequential manner to produce a final product or structure. In everyday life, we assemble furniture, appliances, toys (Ben-Shabat et al., 2021; Ragusa et al., 2021; Sener et al., 2022), etc. Despite having manuals, assembly tasks are challenging due to their complexity or unclear instructions[1]. Beyond the household, assembly extends into industries and workplaces. Manufacturers rely heavily on assembly lines and processes to produce a wide array of goods (Kumar et al., 2022; Cicirelli et al., 2022). These processes are meticulously designed to ensure precision, efficiency, and quality.

In real-world settings, making mistakes is a natural part of assembly tasks. Assembly makes for an interesting subject of study due to its inherent complexity; mistakes in ordering, orientation and fastening are common in assembly (Mattsson & Hogler, 2018; Sener et al., 2022). Dealing with the orientation and fastening mistakes necessitates specialized 3D modeling of components and the estimation of their 6D poses, which falls outside the scope of this work.

This work focuses on detecting ordering mistakes. Ordering mistakes can either be stand-alone or they can can have a cascade effect, impacting subsequent steps and leading to the need for disassembly and reassembly. In the video dataset Assembly101 (Sener et al., 2022), adult participants were asked to assemble and disassemble a toy vehicle designed for 4- to 6-year-olds. However, nearly $60\%$ of the sequences showcased at least one mistake and $78.6\%$ of these were ordering mistakes. An ordering mistake example is shown in Fig. 3(a), if the 'roof' is placed on the 'cabin' before the 'speaker' and the 'light', it will be impossible to position them afterwards. Just like for humans, mistakes serve as valuable learning opportunities for intelligent systems, by revealing possibilities for preventative adjustments.

Sener et al. (2022) were the first to introduce the ordering mistake detection task for assembly tasks together with the release of the video dataset Assembly101. Their benchmark formulated the problem as a video classification task and used a neural network to directly predict mistake labels.

---

[1] A quick online search leads to dozens of articles with titles like "31 Pieces of Furniture You Won't Have a Hard Time Assembling" and "The Secret to Assembling IKEA Furniture Without Losing Your Sanity"

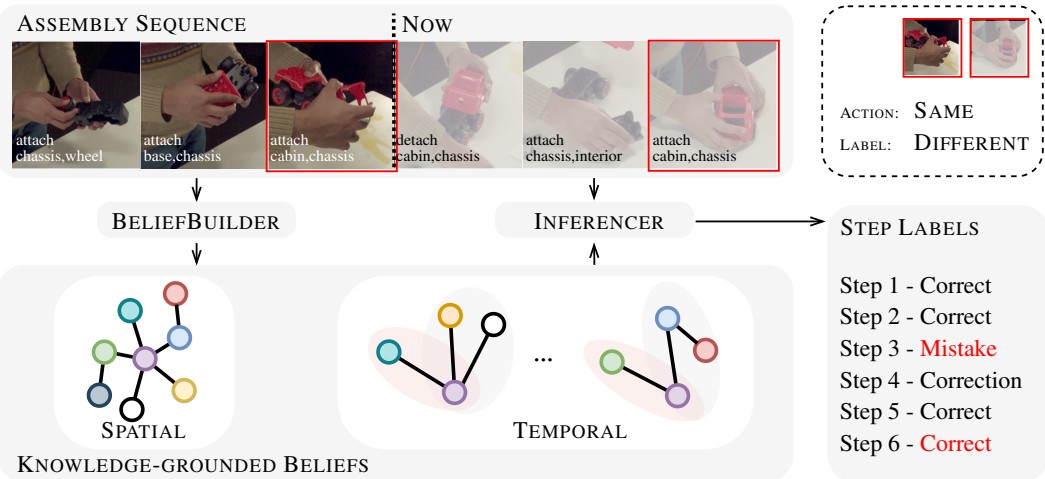

Figure 1: Overview of our online mistake learning and detection system. Given a flow of action sequence, our BELIEFBUILDER progressively augments and revises the knowledge-grounded beliefs at each step. These knowledge are presented as Spatial and Temporal beliefs. Spatial beliefs reveals the structural information of components while temporal beliefs depict the ordering constraints. The INFERENCER leverages the constructed beliefs to make predictions on the action of interest. The mistake relies less on semantic differences and more on the timing of the action. Step 3 & 6 involve the same action but have different mistake labels.

However, this direct mapping from video features to mistake labels conflates the comprehension of action semantics and their temporal relations, making it challenging to interpret the temporal dependencies. We therefore introduce a new set of annotations for Assembly101 that include part information to mitigate the ambiguities. While numerous video datasets exist for procedural activity recognition (Kuehne et al., 2014; Zhukov et al., 2019; Tang et al., 2019), these datasets predominantly showcase successful outcomes and do not endorse any mistakes.

Task graphs (Huang et al., 2019; Logeswaran et al., 2023) can capture the dependencies between the key steps in a task, where Inductive Logic Programming (ILP) algorithms are implemented for the purpose. For example, Sohn et al. (2020) integrates *classification and regression tree* (CART) as the logic induction module to build their task graph with step sequences in a reinforcement setting. It's noteworthy that ILP systems are good at maximizing the objective function with the complete set of data, but struggle to adjust and perform well in dynamic or real-time setups. This makes them less efficient for our mistake detection because of the online detection requirement.

We are motivated to build an adaptive intelligent system that has the ability to learn from mistakes in an online fashion and then identify errors as novel sequences are observed. Specifically, assembly tasks entail a predefined sequence of steps dictated by the components' structure[2]. Furthermore, assembly is a sequential process; each step builds upon previous steps, demanding a specific order of actions. Lastly, assembly actions are reversible and can be undone by disassembly versus procedures like cooking, where actions are irreversible. Inspired by these characteristics, we propose a mistake detection framework presented in Figure 1. The framework consists of two knowledge-grounded beliefs: spatial beliefs, capturing structural relationships between components, and temporal beliefs, encompassing the ordering constraints among action steps. Each of these beliefs attends to a distinct aspect of assembly tasks, as previously outlined. We differentiate between two classes of temporal beliefs due to their different unique error accumulation mechanisms. Additionally, we introduce algorithms designed for the construction of belief sets (BELIEFBUILDER) and the inference (INFERENCER) in an online setup.

To summarize, our main contributions are threefold: **1)** We present two belief sets for the assembly tasks. Spatial beliefs describe relationships between components, while temporal beliefs capture sequencing constraints. These learned beliefs are presented in an explicit form, enabling inspection,

---

[2]We exclude LEGO construction from this work, as it is more free-form than the assembly tasks considered.

comprehension, and validation. Furthermore, our graph representations provide a more intuitive explanation. **2)** We propose a novel mistake detection framework comprising a BELIEFBUILDER and an INFERENCER designed for assembly tasks. The BELIEFBUILDER dynamically constructs the belief sets, while the INFERENCER utilizes these learned beliefs to make predictions and detect mistakes. **3)** We compose a synthetic dataset and enrich the Assembly101 dataset with information about the mistake type and the explicit component connections to facilitate the mistake detection task. We evaluate our method demonstrate its superior capability for mistake detection on both datasets, and additionally show the potential of our approach when integrated with a perception module.

## 2 RELATED WORK

**Task Structure Construction.** Our approach is the first to study ordering mistakes in procedural activities that can handle the flexibility of action order and the variation in approaches of the participants. Soran et al. (2015) tried to detect missing actions for making lattes. In their dataset, 18 of the 41 videos have a purposefully omitted action, *e.g.*, *'steaming milk'*. Soran et al. (2015) model the dependencies of latte-making actions with a directed graph and learn the graph from the complete sequences. Missing actions, however, are not identified until the entire sequence is completed. This method is not generalized to detect assembly ordering mistakes in Assembly 101 since it can only identify the missing steps in a fixed order. Sohn et al. (2020) establish task preconditions using a off-the-shelf inductive logic programming (ILP) module. The same ILP technique has also been employed (Logeswaran et al., 2023) for constructing task graphs solely from textual inputs.

**Mistake vs Anomaly.** Detecting anomalies and unintentional actions (Epstein et al., 2020; Sultani et al., 2018; Chakravarthy et al., 2022; Zatsarynna et al., 2022), especially in temporal sequences, share a similar thread with the objectives of our research. These tasks, involve the identification of actions that deviate from their intended course. However, they distinguish from procedural mistakes in this context, as they exhibit distinct characteristics. Anomalies are actions that stand out due to their unintended nature. For instance, a person who was walking suddenly falls to the ground. The unintentional is discernible by the semantics of falling, as it is not a typical part of walking. Instead, these errors are usually less relevant to semantics and more dependent on the temporal context. In contrast, procedural mistakes in assembly, such as placing parts in the wrong order, typically lack significant semantic implications. These errors are contingent upon the precise assembly steps. This unique distinction highlights the significance of exploring mistake detection in assembly tasks.

## 3 THE APPROACH

### 3.1 PROBLEM SETUP

Consider the assembly of item $\mathbf{X}$ with a component set $\mathbf{P}$. Now consider a collection of $N$ sequences, $\mathbf{S} = \{s_n\}_{n=1}^N$, of people assembling $\mathbf{X}$, and the corresponding labels $\mathbf{Y} = \{y_n\}_{n=1}^N$. Each sequence $s = \{v(i,j)^t\}_{t=1}^T$ has $T$ steps, where $v \in \{attach, detach\}$ denotes the 'verb', and $i, j$ are commutable interacting components, *i.e.*, $i, j \in \mathbf{P}, (i,j) \equiv (j,i)$. The step-wise mistake label is denoted as $y^t \in \{A_{ij}^t, \neg A_{ij}^t, D_{ij}^t, \neg D_{ij}^t\}$, where $A_{ij}^t$ denotes that the step $attach(i,j)$ is correct in sequence and $\neg A_{ij}^t$ is a mistake. Similarly, $D_{ij}^t$ indicates that the step $detach(i,j)$ is expected as a correction of the preceding mistake in time $(\neg A_{ij}^{t'}, t' < t)$ and $\neg D_{ij}^t$ when it is a mistake of unnecessary operation, *e.g.*, taking apart correctly assembled parts.

We further define an episodic context for each sequence $M_n^t = \{y_{t'}\}_{t'=1}^t$ to store the collective steps executed up to time $t$. Due to our online setup of the mistake detection task where the prediction is for the current action, we simplify the notations by omitting $t$. The mistake detection task is then to infer the mistake label $\hat{y}$ for each step $v(i,j)$ in a sequence.

### 3.2 SPATIAL BELIEFS $\mathcal{S}$

In assembly, the component structures dictate the permissible action space. The structural information governs the feasibility of an assembly step and the number of actions required for successful completion. For instance, when assembling toys, a 'roof' component is attached to the 'cabin', and the 'wheels' are affixed to the 'chassis'. As such, we define the spatial beliefs $\mathcal{S}$ to consolidate the

components' structure. More specifically, given components $(i, j)$, the SPATIAL$(i, j, y_s)$ finds the assignment of $\hat{y}_s \in \{A_{ij}, \neg A_{ij}\}$ such that the following formula evaluates to True:

$\hat{y}_s \leftarrow$ SPATIAL$(i, j, y_s)$ :

$$(i, j) \in \mathcal{S} \iff y_s. \tag{1}$$

The formula in Eq. 1 indicates that only $(i, j)$ pairs that conform to the structural constraints belonging to $\mathcal{S}$, is feasible; this is given by definition as $A_{ij}$. The attempt to attach a pair $(i, j)$ that do not fit together, *i.e.*, excluded from $\mathcal{S}$, is a mistake; this is given by definition as $\neg A_{ij}$. Additionally, $\mathcal{S}$ verifies the completion of the assembly task with the following rule:

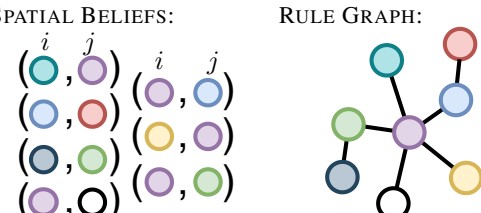

COMPLETED$(M)$ :

$$\forall_{(i,j) \in \mathcal{S}} A_{ij} \in M \iff \text{TRUE}. \tag{2}$$

Figure 2: Spatial beliefs as a set of part pairs (left) and a graph (right).

The rule in Eq. 2 yields a True outcome only when all interconnected components $(i, j) \in \mathcal{S}$ have been successfully assembled by existing steps ($A_{ij} \in M$).

**Graph Interpretation.** The spatial beliefs can be visualized as a graph (Fig. 2), where nodes represent components and edges represent feasible attachments. Completion occurs when the episodic context $M$ fully traverses the graph.

### 3.3 TEMPORAL BELIEFS $\mathcal{T}$

The spatial beliefs verify attachment feasibility without enforcing temporal ordering; they cannot indicate ordering mistakes. As such, we also establish a set of temporal beliefs $\mathcal{T}$ based on observed mistakes during training.

We denote for a *focal* component pair $(i, j)$ its precondition set $\mathcal{D}_{ij}$, a collection of pairs that focal action $attach(i, j)$ relies. Our temporal belief is defined as the union of focal pair and the precondition set, *i.e.*, $\mathcal{T}_{ij} := (i, j) \cup \mathcal{D}_{ij}$. For example, in Fig. 3(a), the focal pair is (roof,cabin), while $\mathcal{D}_{ij}$ consists of (light,cabin) and (speaker,cabin). For a focal action, *e.g.*, $attach(i, j)$, we write as TEMPORAL$(i, j, M, y_t)$ the function that finds the assignment of $\hat{y}_t \in \{A_{ij}, \neg A_{ij}\}$ conforming to the provided formula:

$\hat{y}_t \leftarrow$ TEMPORAL$(i, j, M, y_t)$ :

$$\forall_{(i',j') \in \mathcal{D}_{ij}} A_{i'j'} \in M \iff y_t. \tag{3}$$

The formula in Eq. 3 checks if all precondition pairs from $\mathcal{D}_{ij}$ are assembled before the current action $attach(i, j)$. The focal action is deemed correct only when all its precondition actions are correct.

**Error Accumulation.** Incorrect focal actions in context ($\neg A_{ij} \in M$) may propagate errors to its associated actions within $\mathcal{D}_{ij}$. These errors are referred to as accumulated mistakes. There are two distinct types of error accumulations, based on the transitivity within $\mathcal{T}_{ij}$: *transitive* (represented as $\mathcal{T}_{ij}$) and *intransitive* (denoted as $\neg \mathcal{T}_{ij}$). A transitive belief signifies that any precondition action after the focal is a mistake. For example, in Fig. 3(a), once (roof, cabin) is attached, attaching (light, cabin) and/or (speaker, cabin) will fail and are mistakes. Suppose the pair $(i, j)$ is precondition pair for the focal pair $(i', j')$, meaning $(i, j) \in \mathcal{D}_{i'j'}$. We write the function $\mathcal{T}_{i'j'}(i, j, M)$ with the following rule to infer the label $\hat{y}_t$:

$\hat{y}_t \leftarrow \mathcal{T}_{i'j'}(i, j, M)$ :

$$(i, j) \in \mathcal{D}_{i'j'} \wedge \neg A_{i'j'} \in M \iff \neg y_t. \tag{4}$$

On the other hand, an intransitive belief suggests that if the focal is incorrect, the execution of any of its preconditions is deemed correct, except for the final one. To illustrate, in Fig. 3(b), attach (base, chassis) as a first step would be a mistake according to Eq. 3 because its preconditions are not performed. While the next step attach (cabin, interior) would be considered correct, whereas further attachment of (interior, chassis) would be deemed a mistake. Conversely, should (interior, chassis)

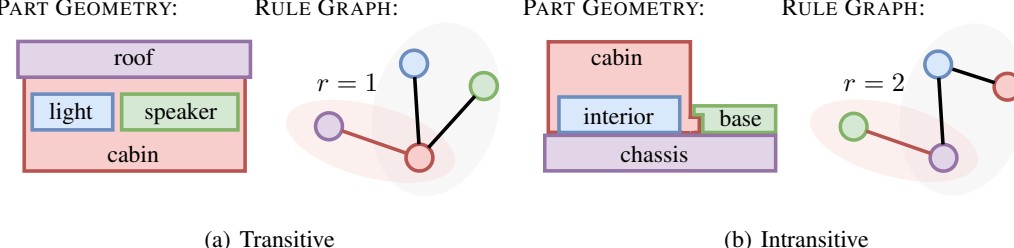

PART GEOMETRY:      RULE GRAPH:      PART GEOMETRY:      RULE GRAPH:

(a) Transitive                 (b) Intransitive

Figure 3: Transitivity of temporal beliefs. The part geometry on the left imposes the ordering constraints. and its graph representation (right). Focal action (the red edge) is reliant on the completion of the rest (black edges in grey group). Transitive and intransitive rules differ in the radius $r$ of their graphs. For transitive rules ($r = 1$), any actions from the dependency set (black edges) will be mistakes after focal action is performed. While, for intransitive rules ($r > 1$), only the last action in black edges that completes the rule graph will be considered mistake.

occurs prior to (cabin, interior), their mistake labels will be swapped. We enforce the following rule in $\neg \mathcal{T}_{i'j'}(i, j, M)$ to obtain the label assignment $\hat{y}_t$:

$$\hat{y}_t \leftarrow \neg \mathcal{T}_{i'j'}(i, j, M):$$

$$\neg A_{i'j'} \in M \wedge (\forall_{(i'',j'') \neq (i,j) \in \mathcal{D}_{i'j'}} A_{i''j''} \in M) \iff \neg y_t. \tag{5}$$

Combining both transitive and intransitive belief, we have the following to make inference for the actions that appear in any precondition sets:

$$\text{PRECONDITION}(i, j, M) \mapsto (\mathcal{T}_{i'j'} \wedge \mathcal{T}_{i'j'}(i, j, M)) \vee (\neg \mathcal{T}_{i'j'} \wedge \neg \mathcal{T}_{i'j'}(i, j, M)). \tag{6}$$

**Graph Interpretation.** Our temporal rules can also be represented as graphs. In their graph representation, the rule's transitivity is determined by its radius. In conventional terms, the radius of a graph is defined as $r = \min_{u \in V} \max_{v \in V} d(u, v)$, where $d(u, v)$ represents the geodesic distance or shortest-path distance between two nodes $u$ and $v$ in graph $V$. Graphs for *Transitive* rules have a fixed radius $r = 1$, while *intransitive* rule graphs have a larger radius of $r > 1$. An illustration is shown in Fig. 3. The correctness of the focal action (indicated by the red edge) relies on traversing the dark edges. It's also possible to consider a hybrid version of these two cases; additional details on this hybrid case are provided in the Supplementary Materials.

### 3.4 BELIEFBUILDER AND INFERENCER

Our attention now shifts to the creation and inference processes of the aforementioned sets of beliefs, facilitated by the introduction of two components: BELIEFBUILDER and INFERENCER. BELIEFBUILDER continuously enhances and revises the belief sets as it parses more streaming sequential action inputs. During inference, the INFERENCER leverages the belief sets to anticipate the output label $\hat{y}$ associated with the observed action, *i.e.*, as correct, mistake or correction.

**BELIEFBUILDER.** The spatial and temporal beliefs are initialized as being agnostic to the task, *i.e.*, set to an empty set $\mathcal{S} = \varnothing, \mathcal{T} = \varnothing$. BELIEFBUILDER proceeds to construct and continuously revise both belief sets as more assembly sequences are observed. Each sequencing error unveils a temporal belief, hence, *every mistake counts in assembly sequences*. For any mistake instance $\neg A_{ij}$, its mistake context $\text{CONTEXT}(i, j, M)$ invariably contains its preconditions. The mistake context, denoted as the set of correct actions occurring between the mistake fix ($D_{ij}$) and its correct execution ($A_{ij}$), *i.e.*, $\text{CONTEXT}(i, j, M) = \{(i', j') | t_{D_{ij}} < t_{A_{i'j'}} < t_{A_{ij}}, A_{i'j'} \in M\}$. The precondition set $\mathcal{D}_{ij}$ for $\mathcal{T}_{ij}$ is updated with the following:

$$\mathcal{D}_{ij} \leftarrow [\mathcal{D}_{ij}] \cap [\text{CONTEXT}(i, j, M)], \tag{7}$$

---

**Algorithm 1** Belief building Step

**function** BELIEFBUILDER($M, i, j, y, \mathcal{C}$)
  **switch** $y$
    **case** $A_{ij}$
      $\mathcal{S} \leftarrow \mathcal{S} \cup (i,j)$
      $\mathcal{C}_{ij} \leftarrow [\mathcal{C}_{ij}] \cap \text{PRECEDES}(i,j,M)$       ▷ Eq. 8
      **if** $D_{ij} \in M$
        $\mathcal{D}_{ij} \leftarrow [\mathcal{D}_{ij}] \cap [\text{CONTEXT}(i,j,M)] \cap [\mathcal{C}_{ij}]$ ▷ Eq. 9
        $\mathcal{T}_{ij} \leftarrow \text{CONNECT}(\mathcal{T}_{ij}, M, \mathcal{S})$       ▷ Eq. 10
        POP($M, \neg A_{ij}$), POP($M, D_{ij}$)
      PUSH($M, y$)
    **case** $\neg A_{ij}$
      **if** ACCUMULATED($\neg A_{ij}$)
        **for** $\neg A_{i'j'} \in M, \mathcal{S}$
          $\mathcal{D}_{i'j'} \leftarrow \mathcal{D}_{i'j'} \cup (i,j)$       ▷ Eq. 11
      PUSH($M, y$)
    **case** $\neg D_{ij}$
      POP($M, A_{ij}$)
    **case** $D_{ij}$
      PUSH($M, y$)

---

**Algorithm 2** Inference Step

**function** INFERENCER($v, i, j, M$)
  **switch** $v$
    **case** 'attach'
      $\hat{y} \leftarrow \text{ATTACH}(i,j,M)$ ▷ Eq. 12
      PUSH($M, y$)
    **case** 'detach'
      $\hat{y} \leftarrow \text{DETACH}(i,j,M)$▷ Eq. 13
      **if** $\hat{y} == D_{ij}$
        POP($M, \neg A_{ij}$)
      **else**
        POP($M, A_{ij}$)
  **return** $\hat{y}$

---

the inclusion of $[\Omega]$ is excluded when $\Omega$ is an empty set. While observing ordering mistakes in the sequences is an evident and potent trigger for the builder to update the temporal beliefs, there exists an implicit temporal logic underlying a completely correct assembly sequences as well. For instance, an action $A_{ij}$ does not have temporal dependencies on its subsequent actions. Conversely, its proceeding actions that are correct, $\text{PRECEDES}(i,j,M) = \{(i',j')|A_{i'j'} \in M\}$, constitute a candidate set $\mathcal{C}_{ij}$ wherein $\mathcal{D}_{ij}$ should be included, *i.e.*, $\mathcal{D}_{ij} \subset \mathcal{C}_{ij}$. $\mathcal{C}_{ij}$ is shared across sequences of actions and continuously refined by:

$$\mathcal{C}_{ij} \leftarrow [\mathcal{C}_{ij}] \cap \text{PRECEDES}(i,j,M). \tag{8}$$

Adding $\mathcal{C}$ to Eq. 7 yields:

$$\mathcal{D}_{ij} \leftarrow [\mathcal{D}_{ij}] \cap [\text{CONTEXT}(i,j,M)] \cap [\mathcal{C}_{ij}]. \tag{9}$$

As illustrated in Fig. 5, it is conceivable that the algorithm could overlook precondition pairs for intransitive temporal beliefs because of the flexible labeling of the precondition actions, as explained by logic formula in Equation (5). The would result in a disjoint intransitive belief graph, meaning the absence of specific black edges in Figure 3(b). To address this scenario, we utilize the spatial belief $\mathcal{S}$ to find the actions present in the episodic context $M$ to establish connection between sub-graphs. Specifically,

$$\text{CONNECT}(\mathcal{T}_{ij}, M, \mathcal{S}) \leftarrow \mathcal{D}_{ij} \cup \{(i'j')|(i',j') \in \text{PATH}(\mathcal{S}, \mathcal{T}_{ij}) \wedge A_{i'j'} \in M\}, \tag{10}$$

where $\text{PATH}(\mathcal{S}, \mathcal{T})$ finds the shortest path in $\mathcal{S}$ that completes the rule graph of $\mathcal{T}$. To accommodate accumulated mistake $\neg A_{ij}$, we would add it into the precondition set of any ongoing order mistakes in context, *i.e.*,

$$\mathcal{D}_{i'j'} \leftarrow \mathcal{D}_{i'j'} \cup (i,j)|\neg A_{i'j'} \in M. \tag{11}$$

The BELIEFBUILDER represented in Algorithm 1 is iteratively applied at each step till the sequence end.

**INFERENCER.** The INFERENCER, as presented in Algorithm 2 is also recurrent and generates predictions based on the belief sets and prior decisions. At each step, the INFERENCER estimates for the tuple $(v, i, j)$ the mistake label, with the context of $M$ which contains the history of action labels up to the current step in that sequence. When the action is to attach, *i.e.*, $v = $'attach', multiple inferences by spatial and temporal beliefs, SPATIAL (Equation (1)), TEMPORAL (Equation (3)) and PRECONDITION (Equation (6)) are made simultaneously to determine its label $\hat{y} \in \{A_{ij}, \neg A_{ij}\}$:

$\hat{y} \leftarrow \text{ATTACH}(i,j,M):$

$$\text{SPATIAL}(i,j) \wedge \text{TEMPORAL}(M,i,j) \wedge \text{PRECONDITION}(i,j,M) \iff y. \tag{12}$$

In the case of the action being detach, *i.e.*, $v = $'detach', the label is contingent upon the context $M$. If the attachment of the same component pairs is an existing mistake, $\neg A_{ij} \in M$, the detachment is a considered as a 'correction' ($D_{ij}$); otherwise, it is a 'mistake' involving unnecessary detachment ($\neg D_{ij}$). Formally, with $y \in \{D_{ij}, \neg D_{ij}\}$, we have:

$\hat{y} \leftarrow \text{DETACH}(i, j, M):$

$$\neg A_{ij} \in M \iff y \tag{13}$$

## 4 EXPERIMENTS

### 4.1 SYNTHETIC DATA

We first leveraged the assembly process of an Epicyclic Gear Train (EGT) from the HA4M dataset (Cicirelli et al., 2022) to create a synthetic sequences with mistakes. HA4M records the assembly of an EGT with provided instructions. We transformed their original 12 actions (verb, noun) to 8 actions (verb, this, that) accounting for component interactions. We manually identify action dependencies, two transitive and one intransitive, by inspecting the geometric constraints of the parts. Subsequently, we generate a synthetic dataset with 20 sequences, 10 of which contain mistakes and 10 are correct. (Refer to Supplementary Material for more detail.)

**Baseline.** We adopt the logic induction approach used by Sohn et al. (2020) as the baseline. We parse the full set of sequences to generate training data for Sohn et al. (2020) as the ideal input while our approach is applied in a streaming fashion.

**Metric.** Similar to Sohn et al. (2020), we evaluate the performance by measuring the agreement between predicted and the ground-truth preconditions for all possible assignments of input. Our metrics include average accuracy (Acc), average precision and average recall.

**Performance.** We conduct a performance analysis by varying the amount of synthetic data accessed by the model and present the results in Table 1. As expected, both approach enjoy a performance boost when more data are input for learning. In both of cases, our approach consistently outperforms Sohn et al. (2020) by a large margin of $> 10\%$ in Acc. Moreover, with only 50% of data, our approach has a recall of 81.2%, sur-

Table 1: Performance of learned rules with all possible input assignments.

|  | Data | Acc | precision | recall |
|---|---|---|---|---|
| Sohn et al. (2020) | 50% | 64.7 | 77.3 | 72.9 |
| Ours |  | 76.1 | 83.9 | 81.2 |
| Sohn et al. (2020) | 100% | 67.6 | 79.8 | 77.0 |
| Ours |  | 83.2 | 89.4 | 85.7 |

passing Sohn et al. (2020) with 100% access of data (77.0%). This is likely due to the challenge for the logic module in Sohn et al. (2020) to generalize and handle the mistakes causes by the intransitive belief, which our approach is capable of learning.

### 4.2 REAL-WORLD DATA

While Assembly101 (Sener et al., 2022) is the sole real-world assembly dataset containing ordering mistakes, their form of annotation '(verb, noun)' lacks the information of the interacting components and can lead to ambiguities. Therefore, we create a new set of annotations for component interactions, represented as (verb, this, that). We hope this form of annotation will inspire further exploration of order mistake detection and even can be helpful for the modeling the object orientation and positioning. Statistics are provided in the Supplementary Material. Coarsely speaking, there are three classes for the mistake detection task: *'correct'* (A), *'mistake'* (B) and *'correction'* (C), where the *'correction'* is a step made to rectify the *'mistake'*. The mistakes can be further classified based on four causes. The most straightforward type is the generic ordering mistake (B1). Accumulated mistakes (B2) are cascaded on generic order mistakes. Another type of mistake is the unnecessary detachment (B3) of correctly assembled parts. Misorientation mistakes (B4) happen when a part is placed in the wrong orientation, such as a reversed cabin. This involves 3D perception and modeling of toy parts, which is beyond the scope of this work.

Table 2: Performance comparison with coarse mistake labels.

| | AR Acc | mistake | | correction | | correct | | Acc | F1 |
|---|---|---|---|---|---|---|---|---|---|
| | | recall | prec. | recall | prec. | recall | prec. | | |
| TempAgg | 100 (GT) | 37.1 | 52.8 | 46.5 | 42.7 | 94.4 | 76.2 | 76.9 | 57.4 |
| LSTM | 100 (GT) | 34.6 | 56.3 | 42.9 | 48.6 | 98.5 | 89.2 | 81.7 | 63.6 |
| Sohn et al. (2020) | 100 (GT) | 65.8 | 59.3 | 23.3 | 57.1 | 94.5 | 91.9 | 83.3 | 62.9 |
| Ours | 100 (GT) | 78.2 | 68.3 | 51.7 | 86.7 | 95.0 | 94.8 | 88.5 | 77.5 |
| Gains | | +12.4 | +9.0 | +28.4 | +29.6 | +0.5 | +2.9 | +5.2 | +14.6 |
| Ours | 86.4 | 42.3 | 53.2 | 40.2 | 55.6 | 73.1 | 84.6 | 74.3 | 56.5 |

**Splits.** In Assembly101, there are a total of 328 distinct action sequences constructing 101 different toys. To create our data splits, we randomly sample one action sequence for testing and use the remaining sequences as the training data for each toy. This process is repeated four times to obtain four splits; we report results averaged over the four splits.

**Evaluation Metric.** We report per class recall and precision, Acc and mean F1 scores over all classes.

## 4.3 EXPERIMENTS

**Baselines.** Following the synthetic experiment, we also use the logic module from (Sohn et al., 2020) as our baseline for comparison. We further compare to two data-driven approaches, *i.e.*, LSTM and TempAgg (Sener et al., 2020), by treating the mistake detection as a sequence-to-sequence task. We design the LSTM (Hochreiter & Schmidhuber, 1997) with four hidden layers, and each hidden layer size set to 256. We train the LSTM with a learning rate of $1e^{-3}$ for 100 epochs. While for the TempAgg, we follow Sener et al. (2022) and train the model for 15 epochs. As inputs, each step in the sequence is represented as its one-hot action feature vector and the action sequences are truncated or padded to a fixed length of 60.

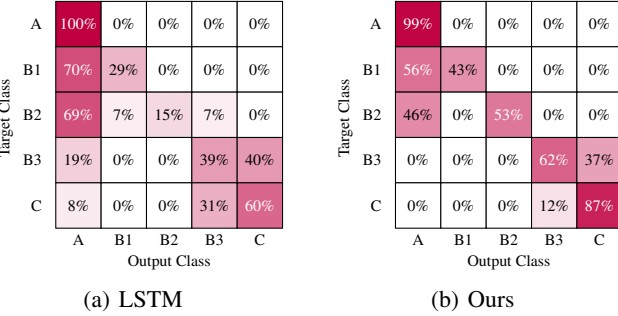

(a) LSTM    (b) Ours

Figure 4: Confusion matrix comparison on the fine mistake labels.

**Coarse mistake detection.** We report the results on Assembly101 dataset with the coarse mistake labels for different approaches in Table. 2. LSTM slightly outperforms TempAgg at the Acc ($+4.8\%$) and F1 ($+6.2\%$) scores. Such a performance gap mainly results from LSTM's boost in the 'correct' class. Our method achieves a high recall of $78.2\%$ on the 'mistake' class, which doubles that ($37.1\%$) of the TempAgg. In the meantime, we are $12.4\%$ and $9.0\%$ higher than (Sohn et al., 2020) in mistake recall and precision, respectively. For other classes, our approach also demonstrates higher performance but with a small gap compared to (Sohn et al., 2020) on the 'correct' class. When evaluated across the classes, ours is the best in both Acc ($88.5\%$) and F1 ($77.5\%$), showing its strong ability to capture the ordering dynamics in assembly sequences.

**Visual Integration.** Assembly101 (Sener et al., 2022) is a challenging dataset for action recognition. Fine-tuned TSM model (Lin et al., 2019) achieves only around 30% accuracy. To show the potential to work with visual perception, we introduce an intermediate scenario in which verbs (attach, detach) are predicted by the fine-tuned TSM module (86.4% in accuracy) with ground truth object parts provided. The result in Table 2 (last row) indicates a decrease of performance, 35.9% in recall while

16.1% in precision for mistake. But this variant still shows higher or comparable performance in detecting mistakes compared to LSTM and TempAgg that use ground truth labels as input.

**Fine mistake detection.** Fig 4 compares confusion matrices for the LSTM and our approach on fine-grained mistakes. As it is shown in Fig 4(a), the LSTM model confuses most of the ordering mistakes (on B1, B2, and B3) with the correct (A) class. This is likely due to the significant imbalance ratio between each fine mistake class and the correct class. However, it also shows that LSTM picks up some knowledge that a detach action can either be a 'mistake' or a 'correction' (see bottom right corner of confusion matrix). Overall, our approach (Fig 4(b)) is better at detecting the ordering mistakes with higher scores along the diagonal.

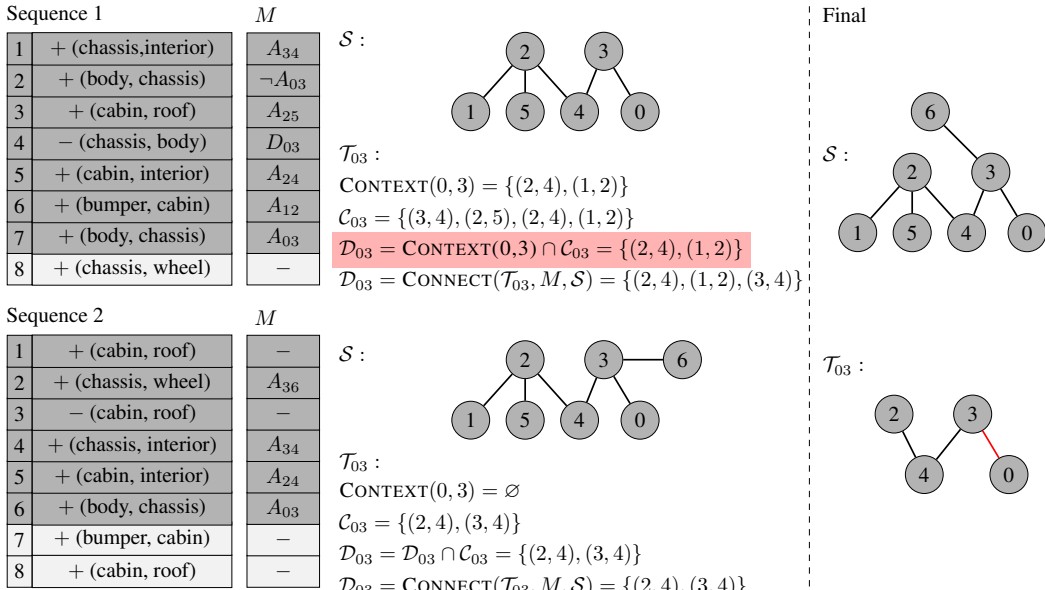

Figure 5: A running example of our BELIEFBUILDER. Upper part shows the algorithm state at the 7-th step in Sequence 1. Within the precondition set $\mathcal{D}_{03}$, which is highlighted in pink, one precondition pair $(3, 4)$ is absent due to its positioning outside the mistake context. Additionally, it includes an extra pair $(1, 2)$ introduced by Step 6. The missing pair is subsequently retrieved through the CONNECT operation, while the extra pair is eliminated when the RULEBUILDER encounters the 6-th step in Sequence 2 in the lower part. The right side exhibits the final spatial and intransitive temporal beliefs discovered by our framework.

**Spatial and temporal beliefs.** We show an example of the beliefs building in Fig. 5. The builder processes the action steps from two sequences progressively, and build beliefs accordingly. After parsing two sequences, a intransitive type of temporal belief is discovered. We additionally run our approach on the full set of sequences on Assembly101 dataset and find 54 temporal beliefs, among which, 28 are transitive and 26 are intransitive. Our empirical observation indicates a strong alignment between the temporal rules obtained and the real-world geometric constraints of certain parts.

## 5 CONCLUSION

This work addresses the challenge of identifying ordering mistakes in assembly tasks. Accordingly, we propose two types of knowledge-grounded belief sets, leveraging the unique characteristics of assembly tasks. The first set captures the spatial arrangement of components, while the other represents the ordering constraints during assembly. These belief sets are generated in a sequence stream with our proposed BELIEFBUILDER and are employed by the INFERENCER to detect mistakes within assembly sequences. Our approach delivers promising results in detecting ordering mistake, consistently surpasses other methods on both synthetic and real-world datasets.

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
