# Supplementary Material for Every Mistake Counts: Spatial and Temporal Beliefs for Mistake Detection in Assembly Tasks

## 1 Part-to-part Annotations Assembly101

The action labels from Sener et al. (2022) are likely to result in ambiguity and inconsistency for the mistake detection task, which is extremely sensitive to the action annotations. The example is shown in Fig. 1. All three cases in Fig. 1 are labeled with the same action label 'attach water tank' by Sener et al. (2022). However, they are fundamentally distinct toy assembly operations that can be distinguished by our part-to-part annotations. Note that Fig. 1(b) is also a self-looped action since 'water tank' is connected to itself.

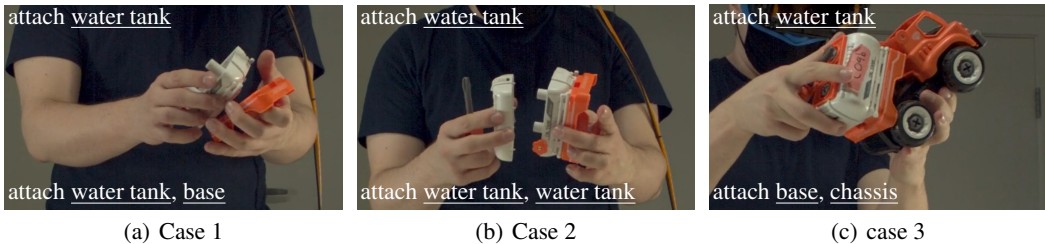

| (a) Case 1 | (b) Case 2 | (c) case 3 |

Figure 1: The coarse action annotation for one video instance from Sener et al. (2022) contains ambiguity. All three instances are labeled as 'attach water tank' (the top line). While (a) connects the 'water tank' to the 'base', b) connects the 'water tank' to itself and c) connects 'base' to 'chassis'. Our part-to-part annotation (the bottom line) provides more precise information.

**Self-looped and repetitive actions.** Certain toy parts from Assembly101 can be further split into two halves, *e.g.*, 'chassis part' and 'water tank part'. For simplicity, we keep a consistent level of annotation and do not consider the subpart level and annotate them as 'attach chassis, chassis'. In this sense, any composition of subparts is a self-looped action. Due to the geometric symmetry of the toys, it is common for assembly sequences to involve repetitive steps. For example, four wheels can be attached at different sequential locations; they are annotated whenever they occur, regardless of their number of occurrences.

Table 1: Six types of mistakes in Assembly101. Misorientation shown in grey as it is beyond the scope of our work.

| Verb | Coarse | # of samples | Remark | Fine | # of samples |
|------|--------|--------------|--------|------|--------------|
| attach | correct | 2927 | correct step | A | 2927 |
| | mistake | 332 | generic order | B1 | 128 |
| | | | accumulated | B2 | 51 |
| | | | misorientation | B4 | 153 |
| detach | mistake | 382 | unnecessary | B3 | 382 |
| | correction | 330 | correction | C | 330 |

**Mistake types.** In Table 1, we provide a breakdown of the statistics of mistakes made by participants during Assembly101. The mistake detection task encompasses three distinct classes: 'correct,' 'mistake,' and 'correction.' Specifically, 'correct' steps (labeled as A) constitute 73.7% of the total, 'mistake' steps (labeled as B) make up 18%, and 'correction' steps (labeled as C) comprise 8.3% of the overall dataset.

The mistakes can be categorized into four distinct types based on their underlying causes. Generic ordering mistakes, labelled as B1, account for 18% of all mistakes. Accumulated mistakes, identified as B2, are the least common, comprising only 7% of the total mistakes, following generic ordering mistakes. The category of unnecessary detachment of correctly assembled parts, labeled as B3, constitutes the majority of mistakes at 54%. Lastly, misorientation mistakes, classified as B4, which involve the incorrect placement of a part, such as a reversed cabin, make up 21% of the total mistakes labels.

## 2 SYNTHETIC DATASET FROM HA4M

In order to demonstrate the generalizability of our model across different assembly tasks, we employed the assembly procedure of an industry item, Epicyclic Gear Train (EGT), from the HA4M dataset (Cicirelli et al., 2022). The HA4M dataset records the assembly of an EGT along with accompanying instructions. In line with these provided sequences, we converted their original actions (comprising a verb and noun) into actions involving component interactions (verb, this, that). We identified 8 actions for successfully assembling an EGT: *1: attach planet gear to planet gear bearing*, *2: attach planet gear bearing to carrier*, *3: attach carrier shaft to carrier*, *4: attach ring bear to sun shaft*, *5: attach sun gear bearing to ring bear*, *6: attach sun gear bearing to sun gear*, *7: attach carrier shaft to sun shaft*, *8: attach cover to ring bear* as show in Figure 2. This figure illustrates the components and geometric constraints of the parts of an EGT, we manually identify two transitive beliefs and one intransitive. Based on the belief, we generated a synthetic dataset with 20 sequences, 10 of which contain mistakes and 10 are correct.

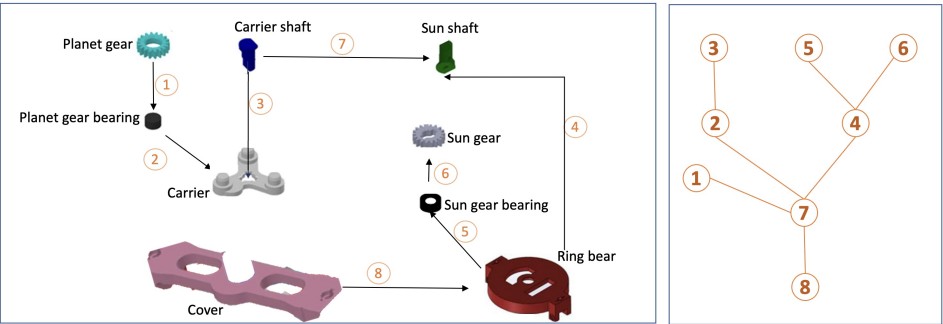

Figure 2: Part connection graph (left) and task graph (right) for EGT.

## 3 HYBRID TEMPORAL BELIEF

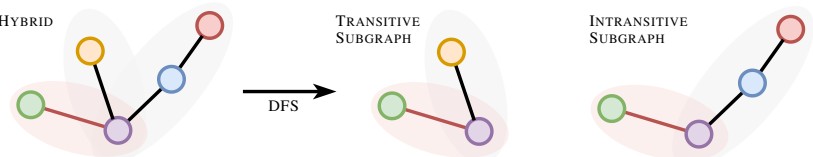

Figure 3: Hybrid temporal belief. The red edge is dependent on the remaining black edges. The hybrid belief can be decomposed into *transitive* and *intransitive* subgraphs by performing a DFS on the graph with an anchor object as root, .

The hybrid temporal beliefs are depicted in Fig. 3. Similar to both *transitive* and *intransitive* temporal rules, the anchor action (red edge) will only be a correct action once the remaining actions (the black edges in the graph) have been completed. While the actions in the precondition set $\mathcal{D}_{ij}$ will apply different rules according to the transitivity of the subgraph they belong. The hybrid belief can be decomposed using depth-first search (DFS) with the anchor object as the root node. Denoting them as $\mathcal{T}_{ij}^{tr}$ and $\mathcal{T}_{ij}^{in}$, we write the following:

$\hat{y} \leftarrow \mathcal{T}_{ij}^{tr}(M)$ :

$$\forall_{i'j' \in \mathcal{D}_{ij}^{tr}} \neg A_{ij} \in M \iff \hat{y} \tag{1}$$

and

$\hat{y} \leftarrow \mathcal{T}_{ij}^{in}(M)$:

$$\neg A_{ij} \in M \wedge (\forall_{(i'',j'') \neq (i',j') \in \mathcal{D}_{ij}^{in}} A_{i''j''} \in M) \iff \hat{y} \tag{2}$$

## 4  ACTION PLANNING

In addition to detecting the ordering mistakes in assembly sequences, it is also possible for us to plan and recommend courses of action. Given an action sequence up to $t'$-th step $\mathbf{s} = \{(v,i,j)_t\}_{t=1}^{t'}$, we wish to generate $\mathbf{p} = \{(v,i,j)_t\}_{t=t'}^{T}$ that not only corrects mistakes in the past but also leads to a successful toy assembly. This is accomplished through the interaction between the proposed INFERENCER and an action SAMPLER. The SAMPLER utilizes a sampling pool that is comprised of unperformed actions and corrections to previous mistakes. The episodic memory $M$ is first obtained by parsing the observations $\mathbf{s}$ through the INFERENCER By interpreting $M$, we can create the following action pool POOL($M$):

$$\text{POOL}(M) = \{(A,i,j)|(i,j) \in \mathcal{S}, A_{ij} \notin M\} \cup \{(D,i,j)|\neg A_{ij} \in M\} \tag{3}$$

The planner will first randomly select an action $(v,i,j)$ from the action pool and use the INFERENCER to estimate its mistake label $\hat{y}$. The action will be discarded if the estimated label is a mistake, *i.e.*, $\neg A_{ij}$; otherwise, the sampled action will be added to the planning list $\mathbf{p}$. The sampling operation will repeat until the toy is fully assembled, *i.e.*, COMPLETED($M$) is True. Alg. 1 provides a synopsis of the action planning procedure. Note that although this planner only addresses the existing mistakes in context $M$ and does not permit any mistakes to be included in the planned sequence $\mathbf{p}$, it is possible to extend the planning process to sample potential mistake actions and then automatically address them.

---

**Algorithm 1** Toy Completion Planner

---

1: **procedure** PLANNER($\mathbf{s}$)
2:     $M \leftarrow \varnothing, \mathbf{p} = \varnothing$
3:     **for** $t \in [1:t']$ **do**
4:         $(v,i,j) \leftarrow s[t]$
5:         $\hat{y} \leftarrow$ INFERENCER($M,v,i,j$)
6:     **while** !COMPLETED($M$) **do**
7:         POOL($M$)                                         ▷ equation 3
8:         Randomly Sample $(v,i,j)$ from POOL($M$)
9:         $\hat{y} \leftarrow$ INFERENCER($M,v,i,j$)
10:         **if** $\hat{y} = \neg A_{ij}$ **then**
11:             POP($M, \hat{y}$)
12:             continue
13:         **else**
14:             PUSH($\mathbf{p}, (v,i,j)$)
15:     **return** $\mathbf{p}$

---