# OpenReview forum: "Every Mistake Counts: Spatial and Temporal Beliefs for Mistake Detection in Assembly Tasks"
_ICLR.cc/2024/Conference — Submitted to ICLR 2024_

### Official Review · Reviewer_6neE · 2023-10-21

**Soundness:** 2 fair
**Presentation:** 1 poor
**Contribution:** 3 good
**Rating:** 5
**Confidence:** 3

**Summary:**

The proposed paper introduces an approach for identifying ordering mistakes in sequential assembly tasks by leveraging knowledge-grounded belief sets. Belief sets are defined to either be spatial or temporal and allow expressing unique characteristics of assembly tasks. The introduced method is validated on synthetic and real datasets.

**Strengths:**

- By focusing on solving ordering mistakes of assembly tasks, this paper addresses an important and open research problem toward solving assembly tasks.
- Experiments are based on synthetic as well as on real data.
- The paper extends Assembly101, an existing dataset, with new labels (although it is not clear if the authors are willing to share their labels and to contribute to the existing work).
- The proposed framework of belief construction (BeliefBuilder) and belief learning (Inferences) appears technically sound and novel.

**Weaknesses:**

- The paper does not discuss any limitations.
- The paper lacks a better exposition (text quality, mathematical notation, figures), which makes it difficult to fully assess the contribution.
- Missing ablation studies.

**Questions:**

- Section 3.2: What is meant by 'given by definition as A_{ij}'?
- The in-text description of Figure 2 refers to 'edges', but no edges are shown in Figure 2 (left). Does this mean the graph has no edges?
- Section 3.2: The text states that 'completion occurs when the episodic context M fully traverses the graph'. Does this refer to the rule graph?
- What is a 'focal component pair'?
- The mathematical notation appears convoluted and more difficult to follow that it seems necessary. After
reading Section 3.2 and 3.3, it was not clear how Spatial and Temporal beliefs are defined.
- Given the quality of the exposition, I have the impression that this paper was rushed and that the text needs to be refined prior to publishing this work.
- Section 4.3: What are 'fine-grained' mistakes?
- To assess the performance of the Inferences vs the BeliefBuilder, I would appreciate ablation studies.

**Details Of Ethics Concerns:**

No ethical concerns regarding this submission.

---

> ### Author Response · Authors · 2023-11-23
>
> We appreciate the reviewer's recognition of our paper's significance in addressing ordering mistakes in assembly tasks and kind comments on our data, new labels and proposed framework.
>
> ----
>
> **W1. The paper does not discuss any limitations.**
>
> **A1**: Thanks for this feedback! We acknowledge that, in its current form,  our current approach doesn't generalize well to new object parts and is heavily influenced by perception quality. We'll incorporate these points into our revision.
>
> **W2. The paper lacks a better exposition (text quality, mathematical notation, figures), which makes it difficult to fully assess the contribution.**
>
> **A2**: We appreciate the reviewer's feedback and will clarify and refine our contributions in our paper and will enhance its clarity.
>
> **W3. Missing ablation studies.**
>
> **A3**:  Unfortunately, our framework does not allow for various ablations. In Table 2,  we've explored various perception module levels to show the gap between GT and computer vision-based perception modules.
>
> ----
>
> **Q1. Section 3.2: What is meant by 'given by definition as $A_{ij}$?**
>
> **A1**: $A_{ij}$ is defined in Sec. 3.1 and indicate that the action of attaching $i$ and $j$ is a correct action.
>
> **Q2. The in-text description of Figure 2 refers to 'edges', but no edges are shown in Figure 2 (left). Does this mean the graph has no edges?**
>
> **A2**: Thanks for bringing this to our attention!  We’ll revise the corresponding text to refer to Figure 2 (right), when discussing edges,
>
> **Q3. Section 3.2: The text states that 'completion occurs when the episodic context M fully traverses the graph'. Does this refer to the rule graph?**
>
> **A3**: Yes, your understanding is correct! It does refer to the rule graph.
>
> **Q4. What is a 'focal component pair'?**
>
> **A4**: The term is to denote the part pairs of interest that the temporal rule is designed for as described in Sec3.3 (the second paragraph).
>
> **Q5. The mathematical notation appears convoluted and more difficult to follow that it seems necessary. After reading Section 3.2 and 3.3, it was not clear how Spatial and Temporal beliefs are defined.**
>
> **A5**: Sec 3.2 explains that spatial belief is defined as the allowed connection between parts, which is also considered as the edge of the rule graph. Sec 3.3 essentially talks about two types of temporal beliefs and how they can be used to infer ordering mistakes.
>
> **Q6. Given the quality of the exposition, I have the impression that this paper was rushed and that the text needs to be refined prior to publishing this work.**
>
> **A6**: We thank the reviewer for the comment, and we are actively working on revising the text to address the concern and enhance the overall quality.
>
>
>
>
> **Q7. Section 4.3: What are 'fine-grained' mistakes?**
>
> **A7**: The coarse mistakes only care about mistakes, correct and corrections without accounting for their root causes. Following this, the fine-grained mistakes are introduced in Sec 4.2 (the last paragraph on page 7), which include generic ordering mistakes (B1), accumulated mistakes (B2), unnecessary detachment (B3), and Misorientation mistakes (B4). In  addition, we provided more information on the mistake types in our supplementary.
>
>
> **Q8. To assess the performance of the Inferences vs the BeliefBuilder, I would appreciate ablation studies.**
>
> **A8**: The BeliefBuild essentially builds the rule graph which is later used by the Inferencer during testing only. We therefore cannot compare the Inferencer vs the Beliefbuilder.

---

### Official Review · Reviewer_MJjr · 2023-10-31

**Soundness:** 2 fair
**Presentation:** 2 fair
**Contribution:** 1 poor
**Rating:** 3
**Confidence:** 4

**Summary:**

The paper studies a relatively new and interesting problem of mistake detection in assembly tasks. In particular, the paper deals with the detection of ordering mistakes only using the spatial and temporal knowledge of the part names and their action verbs (attach or detach). Spatial knowledge represents the topological conformity, whereas the temporal knowledge accounts for the sequencing constraints. The paper does not deal with assembly mistakes occurring due to orientation or fastening, and also leaves out the perception problems that need to be addressed for this task. The method uses synthetic dataset to determine the efficacy of their pre-conditions with respect to the state of the art method Sohn et al. The mistake detection of this proposed approach is evaluated on the annotated toy assembly dataset - Assembly 101.

**Strengths:**

The paper presents a timely work that focuses on a new and understudied problem of ordering mistake detection in assembly tasks.
The paper is well written and structured, providing clarity to the reader through good examples in the text, wherever required.

**Weaknesses:**

Trivial task definition: Examining only the errors related to the ordering in assembling tasks using the ground truth annotated object names and action verbs seems trivial, unless the issues related to orientation or fastening and the uncertainties in the object detection, shape, size or pose estimation are taken into consideration.
Perception is ignored : Perception is an essential component of this problem statement. The real-life applicability and deployment seems impractical, unless the method is evaluated properly with a perception module that addresses object detection, shape & pose estimation etc. The results shown in the paper using the TSM perception to predict only the action verbs - attach or detach, shows a drop in recall by 35.9% and precision by 16.1%, highlighting the significance of perception for this problem.
Limitations in the approach : The proposed approach is based on pure logic, set theory and graph representation of the spatial and the temporal relations. The entire pipeline relies on hard-coded logic and does not employ any learning mechanisms, leading to re-initialisation and re-population of the spatial and temporal knowledge, along with the preconditions, each time this method needs to assess a new object assembly. On top of that further improvements in this deterministic pipeline needs to be manually accounted for, hence inclusion of this paper is of little importance to the learning community.
Limited evaluation : The Assembly 101 dataset has only 328 distinct action sequences to construct 101 different toys and out of this only 1 randomly selected sequence is used for testing for each toy and remaining are used to create the spatial & temporal knowledge and the preconditions. The complexity of sequence in assembling a toy is not an apt illustration of the difficulty in assembling an object with numerous parts, involving a large complex sequence, say an electronic gadget, automobile etc. In addition, this dataset is too small in size to effectively represent all the possible scenarios of ordering mistakes that occur in real-life large scale assembly, for example a factory. The criteria of the logic module appears to be tailor made for the toy assembly dataset. Therefore, its performance in a different assembly scenario, with more parts and complex sequences needs to be sufficiently evaluated to establish the generalizability.
Limitations in the logic : In Tab. 1, the proposed method shows a significant improvement over Sohn et al. (2020), but that is likely because of the inclusion of an additional case (intransitive temporal error) in the logic module. Yet, the proposed approach is unable to achieve 100% accuracy, precision or recall, clearly showing gaps in their logic module to determine all the possible cases of preconditions. Similarly in Tab. 2,  the proposed approach performs better than the existing methods, but I am unable to contemplate a reason for its failure to achieve perfect accuracy and F1, except the possibility of missing out on certain scenarios in its logic criterion, despite being a hard coded algorithm.
Minor Weaknesses :
Clumsy Notations : Too many redundant symbols and equations appear in the paper whose purpose could have been simply stated in words.
Clarity in Fig 3 : Edges for both transitive and intransitive graphs have the same color. Especially in the intransitive graph, the “final mistake” edge could have been highlighted differently.
Clarity in Fig 5 : Please state the numerical identity of the assembly parts clearly to make the figure more intelligible

**Questions:**

Is it really difficult to integrate state of the art perception with this pipeline? If so, why? It would be really helpful if you could provide the end-to-end results of this method and state-of-the-art (SOTA) with perception. A real-time video demonstration showing the ordering mistake detection in the assembly task, using only the RGB frame, would be very intriguing.
Is it not possible to create and evaluate this method and SOTA on a larger dataset with different objects, more complex sequences and more parts? If so, what are the difficulties? A larger dataset would alleviate the concerns of generalizability.
If perception is included in this pipeline and you are able to estimate the object pose, how difficult would it be to extend this method to detect orientation errors?

---

> ### Author Response · Authors · 2023-11-23
>
> We appreciate the reviewer's kind words on our well-timed framework for a previously overlooked challenge and our clear and organized presentation.
>
> ----
>
> **Trivial task definition**
>
> **A**: We focus on ordering mistakes in this paper, which comprise 2/3 of overall mistakes in Assembly101 and require temporal reasoning. We do agree with the reviewer that object pose prediction and action recognition are worth investigating as challenging tasks in the computer vision domain. We refrain from shifting our attention to such aspects in this work for the following considerations:
>
> - Object pose prediction [A] primarily focuses on large objects, requiring object meshes for seen classes for some models; Assembly101 lacks these meshes. Additionally, studying orientation issues is a frame-based investigation problem, which is also orthogonal to our temporal reasoning for the ordering mistake detection.
> - Action recognition on the well-studied egocentric dataset Epic-Kitchens reaches only 56% accuracy [B]. Research [C] on adverbs like "tight" modifying the verb "fasten" has been limited to small-scale studies, achieving 71.9% accuracy for just 5 adverbs.
>
> **Perception is ignored**
>
> **A**: Perception is an orthogonal line of work that requires more progress until we can fully realize a vision-based intelligent assistant which can simply observe our setting and respond.
> - On one hand, we utilize GT labels to isolate mistake detection from perception module errors, which provides a unbiased presentation of the ordering mistake detection problem.
> - On the other hand, we did not ignore perception as we integrated the existing action recognition outputs with our model and reported the results in Table 2.
>
> Table 2 shows that our accuracy decreases due to accumulated perception errors, highlighting the need for improved perception to enhance mistake detection.
> By using GT labels, we can continue to advance research on mistake detection without being hindered by limitations in CV-based recognition, enabling progress despite these challenges. This is similar to how QR codes and RFID tags aid robot navigation, providing identifiable markers to tackle the limitations of the perception module.
>
> **Limited evaluation**
>
> **A**: We politely disagree that our approach is tailored for the toy assembly.
> - Our spatial and temporal beliefs are generic, not specific to toy parts; as we tested them on 101 different toys. Through synthetic experiments in our paper, we demonstrate the adaptability of our approach for EGT assembly in industry settings, serving as a proof of concept for its potential to generalize and improve assembly tasks across various domains.
> - Assembly101 is the sole dataset with these annotations. And it encompasses intricate assembly scenarios, averaging 14 steps, making it a valuable resource for studying mistake detection and learning. Even adults assembling children's toys make numerous mistakes, highlighting the complexity of the task and the need for improved learning strategies.
>
> **Limitations in the approach**
>
> **A**: We acknowledge that our approach is based on logic which build the spatial and temproal knowledge, however, we respectfully disagree with the reviewer that the inclusion of our work in ICLR is of less importance. Specifically, our spatial and temproal beliefs can be of great interest to the audience working with robotics, planning.
>
>
> **Limitations in the logic**
>
> **A**: The reason that our approach is not 100% accurate is due to the training data.  Our approach's accuracy is limited by training data constraints. When we only have access to a subset of error-free action sequences, it will hinder our algorithm's ability to handle mistakes during testing. We performed our algorithm over 4 splits to account for this.
>
>
>
> **Minor**: Clumsy Notations, Clarity in Fig. 3 and Fig. 5.
>
> **A**: Thanks for the comment.
> - We do state everything in words, either above or below the equations. We found that the equations provide a precise and concise way to present complex relationships, making the information more unambiguous. They enable precise communication of ideas which can be challenging to convey in words alone.
> - Thanks for pointing this out and we’ll highlight the differences and update in the figures.
>
> [A] BOP Challenge 2022 on Detection, Segmentation and Pose Estimation of Specific Rigid Objects, CVPRW 2023
>
> [B] Training a Large Video Model on a Single Machine in a Day. arXiv preprint arXiv:2309.16669 (2023).
>
> [C] Learning Action Changes by Measuring Verb-Adverb Textual Relationships. in CVPR. 2023.

---

> ### Author Response · Authors · 2023-11-23
>
> **Questions**
>
>
> **Is it really difficult to integrate state of the art perception with this pipeline? If so, why?**
>
> **A**: In this paper, we decouple the perception and reasoning modules. We can utilize the output of the perception module, simply by replacing the GT labels with the output of the perception module which are expected to be the per-frame/segment action labels,  as seen in Table 2.
>
>
> **It would be really helpful if you could provide the end-to-end results of this method and state-of-the-art (SOTA) with perception.**
>
> **A**: We integrate our mistake detection module with the perception module's outputs in Table 2. Notably, when AR Acc is 86.4%, errors in the perception module result in a decrease in mistake recall to 42.3%, highlighting the module's impact.
>
>
> **A real-time video demonstration showing the ordering mistake detection in the assembly task, using only the RGB frame, would be very intriguing.**
>
> **A**: We will provide these for both ground truth labels and perception module outputs.
>
>
> **Is it not possible to create and evaluate this method and SOTA on a larger dataset with different objects, more complex sequences and more parts? If so, what are the difficulties?  A larger dataset would alleviate the concerns of generalizability.**
>
> **A**: The primary challenge in adapting our approach is the scarcity of datasets with mistake labels. Assembly101 is the only existing large dataset providing such labels. To address this, we created the synthetic EGT dataset, which our model performs well on. However, the lack of alternative mistake datasets limits further testing and validation on real data.
> We can in principle adapt our approach to more complex sequences and more parts, which may require to account for the hybrid type of temporal belief as introduced in our supplementary.
>
>
> **If perception is included in this pipeline and you are able to estimate the object pose, how difficult would it be to extend this method to detect orientation errors?**
>
> **A**: Orientation is independent of temporal reasoning and can be addressed as a frame-wise issue through object pose prediction. We expect that with an accurate object pose prediction method, detecting orientation mistakes is feasible.
> To avoid confusion, we are considering renaming our paper to clarify that we focus on detecting ordering mistakes.

---

> > ### Comment · Reviewer_MJjr · 2023-12-02
> > **Response to Authors**
> >
> > I appreciate the authors' efforts in addressing my concerns.
> >
> > **Trivial task definition**
> > - I understand that Assembly 101 does not have object meshes to support [A] but as authors have stated the action recognition on Epic-kitchens reaches only 56% accuracy, it indicates a substantial gap in the state-of-the art perception for addressing these tasks.
> > - I still believe that the claim regarding mistake detection in the paper is very limited, as it exclusively focuses on a narrow subset of mistakes related to ordering in assembly, while disregarding errors associated with orientation or fastening.
> >
> > **Perception is ignored**
> > - I disagree with the authors, given that the output from the perception module serves as the input to the pipeline, any errors in perception will propagate downstream, thereby impacting the accuracy of mistake detection.The results further indicate a considerable drop in recall and precision when perception is included, highlighting the necessity for improvement. A potential improvement lies in remodeling the method to mitigate perception errors, a modification that could lead to significant impact in this domain.
> >
> > **Limited evaluation**
> > - The empirical proof for the generalizability of the spatial and temporal modules needs to be established through testing in assembly tasks involving a diverse array of parts and encompassing extensive, intricate sequences, such as those found in electronic gadgets or automobiles.
> > - Assembly 101 is too small in size to effectively represent all the possible scenarios of ordering mistakes that occur in real-life large scale assembly.
> >
> > **Limitations in the approach**
> > - The need for re-initialisation and re-population of the spatial and temporal knowledge, along with the preconditions, each time this method needs to assess a new object assembly, poses significant limitations. The lack of learning in this method prevents it from capturing the generic intelligence required to detect ordering mistakes in a large variety of assembly tasks without re-initialisation or re-population. Therefore, the paper has limited impact in its current form.
> >
> > **Limitations in the logic**
> > - I understand the method only captures the preconditions from a subset of error-free sequences but this should not be a limiting factor for the performance of this method. Notably, there is no module in the pipeline that captures generic intelligence for ordering detection, limiting its ability to extend the mistake detection for a new object assembly.
> >
> > Owing to these reasons, I want to keep my rating.

---

### Official Review · Reviewer_ugZt · 2023-11-01

**Soundness:** 3 good
**Presentation:** 2 fair
**Contribution:** 2 fair
**Rating:** 5
**Confidence:** 2

**Summary:**

This paper is concerned with identifying ordering mistakes in assembly tasks. The proposed approach uses spatial and temporal beliefs that store structural and ordering constraints respectively. These are then used to infer mistakes in action sequences. The paper evaluates on both synthetic and real-world data.

**Strengths:**

- Interesting subject matter and difficult problem. The paper explains well the non-obvious challenges behind detecting ordering mistakes.

- The modeling of the temporal belief is intuitive and well thought out.

**Weaknesses:**

- Writing needs some improvements. For example, the task needs to be explained better. Especially for readers that are unfamiliar with the assembly ordering mistake task, it is not clear whether the input is a sequence of images or labels representing the actions. I assume it is the latter, since the authors provide a separate Visual Integration experiment towards the end of the paper. Moreover, if the part-to-part annotations and the synthetic dataset are treated as contributions, then more information should be included in the main paper, rather than the supplementary material. Please add a couple examples of the synthetic sequences with mistakes.

- In my understanding, the BeliefBuilder is creating an explicit rule graph that only applies to a specific object and its set of components. Aren't explicit knowledge grounded beliefs very limited in terms of generalization? Wouldn't it be more useful to learn a soft graph with latent representations that could potentially be applied to new objects?

**Questions:**

- What is the performance of the Inferencer on Seen vs Unseen objects?

---

> ### Author Response · Authors · 2023-11-23
>
> We appreciate the reviewer's thoughtful feedback, and thank for recognizing our paper's topic and how challenging it is and our design for temporal belief modeling.
>
> ----
>
> **W1. It is not clear whether the input is a sequence of images or labels representing the actions. Moreover, if the part-to-part annotations and the synthetic dataset are treated as contributions, then more information should be included in the main paper, rather than the supplementary material. Please add a couple examples of the synthetic sequences with mistakes.**
>
> **A1**:
> - Yes, our approach generally takes as input labels and such labels can be obtained from multiple sources, for example, the video annotation, parsed from narration in the video, or the output from a perception module. In this work, we evaluate our approach with both annotation and perception outputs in Table 2.
> -  Thanks for highlighting the synthetic data issues. We'll add the examples to the main paper for revision.
>
>
> **W2. In my understanding, the BeliefBuilder is creating an explicit rule graph that only applies to a specific object and its set of components. Aren't explicit knowledge grounded beliefs very limited in terms of generalization? Wouldn't it be more useful to learn a soft graph with latent representations that could potentially be applied to new objects?**
>
> **A2**: We haven't explicitly tested for generalization, but our model is expected to perform reasonably for shared part pairs like interior-cabin and sound module-roof on different toys. However, for novel object parts across toys, prior knowledge is necessary, e.g., informing the model about the similarity of attaching a dumper vs a container which are usually positioned in the rear part of the toy. One option to get such information could be through leveraging LLMs, so that we can generalize to new objects like unknown containers by recognizing structural similarities aligned with linguistic semantics. We thank the reviewer and will incorporate these discussions into our paper.
>
>
> **Q1. What is the performance of the Inferencer on Seen vs Unseen objects?**
>
> **A3**: As we mentioned above, we focus on seen objects in this paper, but with structural prior knowledge, our rule graphs can be extended to unseen objects.

---

### Official Review · Reviewer_RoBJ · 2023-11-01

**Soundness:** 2 fair
**Presentation:** 1 poor
**Contribution:** 2 fair
**Rating:** 5
**Confidence:** 2

**Summary:**

This paper presents a method for detecting ordering mistakes in assembly tasks. The authors introduce two beliefs (spatial and temporal beliefs) in the method for modeling the part relationships and propose BeliefBuilder/Inferencer for training/test. The method is tested on a synthetic dataset and a real dataset Assembly101.

**Strengths:**

- The proposed spatial and temporal beliefs, and the mechanism for modeling the part assembly ordering, make sense and look valid.
- Empirical results show improved performance over baseline methods.

**Weaknesses:**

- The paper is hard to read and follow. The system is described in a complicated way, though the underlying mechanism is straightforward.
- The paper only tackles ordering assembly mistakes, which is just one type of failures. LEGO construction tasks are also excluded.
- The method avoids perception and reasoning, but instead resorts to human annotations for the parts and the steps, which is an unrealistic simplification of the problem.

**Questions:**

See weakness

---

> ### Author Response · Authors · 2023-11-23
>
> We appreciate the reviewer's positive feedback on the proposed spatial and temporal beliefs and our improved performance over baseline methods.
>
> ----
>
> **W1. The paper is hard to read and follow. The system is described in a complicated way, though the underlying mechanism is straightforward.**
>
> **A1**: Thank you for your comments. Even though the underlying mechanism appears straightforward at first glance, unraveling the less apparent challenges within ordering mistake detection requires a detailed and comprehensive description. We will improve the readability and ease of comprehension to ensure our paper is accessible to a broader audience.
>
> **W2. The paper only tackles ordering assembly mistakes, which is just one type of failures. LEGO construction tasks are also excluded.**
>
> **A2**: We agree that solving LEGO and other assembly mistakes are interesting.  For a scientific paper, however, the primary aim often revolves around delving deep into a specific problem setup rather than addressing the broader spectrum of challenges involved in product development. While the paper might focus narrowly, its insights can contribute to a broader understanding of the field. LEGO and other assembly mistakes are out of scope, as we explain below.
> * The Assembly101 dataset breaks down the mistakes - ordering mistakes comprise 2/3 of the overall mistakes.  The remaining third are 3D placement mistakes.  Solving these mistakes requires 6D pose estimation of the object parts and is a vastly different problem than our focus on temporal reasoning and outside our scope.
> * We exclude LEGO because the underlying problem setting is fundamentally different and we explain this explicitly in our manuscript. LEGO bricks are designed to allow creative freedom to allow open-ended construction.  Without brick semantics, mistakes mainly arise from imprecise attachments, which require tactile feedback that is an entirely different field much less beyond the scope of this work.
>
> **W3. The method avoids perception and reasoning but instead resorts to human annotations for the parts and the steps, which is an unrealistic simplification of the problem.**
>
> **A3**: We respectfully disagree.  Our work simply separates perception and reasoning into two separate components, and two are developed independently.
> * Reasoning: the separation of reasoning from the perception is standard in robotics research [A];  Mistake detection is essential for many applications, and waiting for perfect perception to be solved is not a practical approach for advancing research.
>
> * Perception: This is not our main focus so we rely on existing model (TSM), which we train for our problem setting.  We verify that using perception-based outptus integrates seamlessly in our approach as shown Table 2.  While there are some performance drops due to imperfect perception, we don’t think this takes away from the underlying contributions of our proposed work.
>
>
> [A] Differentiable Parsing and Visual Grounding of Natural Language Instructions for Object Placement. In ICRA, 2023

---

### Meta-Review · Area_Chair_uxRi · 2023-12-10

**Metareview:**

The submission proposes a new method to identify mistakes in assembly tasks.   While reviewers liked the problem and formulation, all four of them were concerned about the presentation and limited scope and evaluation of the submission. Eventually, all of them recommended rejection.  The AC agrees.  Some reviewers in particular raised concerns regarding the lack of a perception module.  The AC  agrees with the authors' note to AC that this submission could make a contribution without tackling the perception problem: in addition to the experiments that the authors have included in Table 2, the authors may also use e.g., motion capture to obtain pseudo-ground-truth object states and test the system's performance in that setting. The authors are encouraged to revise the submission based on these comments for the next venue.

**Justification For Why Not Higher Score:**

Shared concerns about the presentation and limited scope and evaluation of the submission.

**Justification For Why Not Lower Score:**

N/A

---

### Decision · Program_Chairs · 2024-01-16

Reject